# Unveiling Monoterpene Biosynthesis in *Taiwania cryptomerioides* via Functional Characterization

**DOI:** 10.3390/plants10112404

**Published:** 2021-11-08

**Authors:** Li-Ting Ma, Pi-Ling Liu, Yang-Tui Cheng, Tz-Fan Shiu, Fang-Hua Chu

**Affiliations:** School of Forestry and Resource Conservation, National Taiwan University, Taipei 10617, Taiwan; mary2234@gmail.com (L.-T.M.); bad8016479@gmail.com (P.-L.L.); timmy304681@gmail.com (Y.-T.C.); jimmy81513@hotmail.com (T.-F.S.)

**Keywords:** *Taiwania cryptomerioides*, monoterpene synthases, gas chromatography-mass spectrometry (GC-MS)

## Abstract

*Taiwania cryptomerioides* is a monotypic species, and its terpenoid-rich property has been reported in recent years. To uncover monoterpene biosynthesis in *T. cryptomerioides*, this study used transcriptome mining to identify candidates with tentative monoterpene synthase activity. Along with the phylogenetic analysis and in vitro assay, two geraniol synthases (*Tc*TPS13 and *Tc*TPS14), a linalool synthase (*Tc*TPS15), and a β-pinene synthase (*Tc*TPS16), were functionally characterized. Via the comparison of catalytic residues, the Cys/Ser at region 1 might be crucial in determining the formation of α-pinene or β-pinene. In addition, the Cupressaceae monoterpene synthases were phylogenetically clustered together; they are unique and different from those of published conifer species. In summary, this study aimed to uncover the ambiguous monoterpenoid network in *T. cryptomerioide*, which would expand the landscape of monoterpene biosynthesis in Cupressaceae species.

## 1. Introduction

*Taiwania cryptomerioides* Hayata is a monotypic species that has economic and ecological importance in Taiwan. Due to its outstanding durability, the *T. cryptomeriodies* plantation has been ongoing around mountainous areas of Taiwan [1], and it has attracted attention in research on bioactivity and chemical composition [2,3]. Previous studies have reported that terpenoids isolated from *T. cryptomerioides* have great bioactivity, such as antifungal, antimite, and antioxidation activities, and most of them are sesquiterpenoids and diterpenoids, which are extracted from wood [3,4,5,6]. However, the needles of *T. cryptomerioides* contained high amounts of acrylic and cyclic monoterpenoids, such as limonene, pinenes, terpinolenes, and some of their derivatives [7,8,9,10,11,12], but their biosynthesis remained ambiguous.

The monoterpene was composed of two isoprene units, which were condensed by geranyl pyrophosphate (GPP) synthase [13]. Furthermore, GPP was transformed to monoterpenes by a group of terpene syntheses (TPSs), named monoterpene synthase [14]. These enzymes contained DDxxD and NSE/DTE motifs, and they are essential in binding and detaching the pyrophosphate group via metal ions. Once the substrate cation forms, the catalytic residue around the substrate will further deprotonate, hydroxylate, or cyclize the cation until it becomes a stable product [15]. The conformation of the active site is determined by neighboring residues, and it affects the carbon skeleton of the product [16,17,18]. Thus, it is important to functionally characterize monoterpene synthase to better understand the logic behind the diverse chemical structures.

The evolution of monoterpene synthases has been suggested as a rapid and predominantly divergent process [19,20]. Although it can be easily distinguished from sesqui- and diterpene syntheses due to domain loss events [19], it is unlikely applicable to predict the exact product of monoterpene synthases based on sequence similarity. Nevertheless, the product specificity of monoterpene syntheses is determined by catalytic residues around the substrate, and phylogenetically related homologs likely share similar plasticity residues [17,21,22]. Thus, it is a challenge to predict the function of monoterpene synthases among non-model species such as *T. cryptomerioides*, and it requires a complete biochemical characterization of monoterpene syntheses as a template for further research. To address this issue, high-throughput transcriptome sequencing is a reliable tool to discover potential participants in the network of *T. cryptomerioides* monoterpene biosynthesis.

In previous studies, few Cupressaceae-associated clusters were identified in sesquiterpene and diterpene synthases via phylogenetic analysis [23,24,25], and Cupressaceae monoterpene synthases might hypothetically be close in phylogenetic relationships. Compared with diterpene synthases, monoterpene and sesquiterpene synthases evolved much later after losing γ-domains, and the remaining β-α domain exhibited highly conserved organization. However, the diverse catalytic functions of sesqui- and monoterpene synthases suggest that the plasticity residues in these enzymes have been the major cause of divergent evolution, and few are known in gymnosperm species, especially Cupressaceae.

In this study, we identified monoterpene synthases from *T. cryptomerioides* needle and cone transcriptome library and aimed to provide complete information on monoterpene biosynthesis in *T. cryptomeioides* by conducting phylogenetic analysis, an in vitro enzymatic assay, and catalytic residue comparison. The new discovery of a Cupressaceae-specific cluster of monoterpene synthases revealed the phylogenetically and biochemically divergent relationship of conifer monoterpene synthases and unveiled the hidden terpene biosynthetic network, which is crucial to plant defense and ecological communication.

## 2. Results and Discussion

### 2.1. TcTPS Isolation

To identify any potential candidates for monoterpene synthase, we searched previously established *T. cryptomerioide* transcriptome libraries, which include needles, cones, and wood. After conducting BLAST annotation against the NCBI nonredundant (NR) database and removing duplicates, three contigs (IDs 29358, 5958, and 2226) from the cone library and two contigs (IDs 17110 and 10575) from the needle library were identified as candidates (Table 1). Although monoterpene synthase was also observed in the wood transcriptome library, the lengths were shorter than 500 bp and identical to the contigs from needles and cones. This result might be caused by the low abundance of monoterpene synthases in wood, which corresponded to the lower concentration of monoterpenoids in the metabolite profile of *T. cryptomerioides* wood extracts and essential oils in previous studies. Thus, we decided to use contigs from needles and cones for further experiments and renamed them *TcTPS13* (*TF_29358*), *TcTPS14* (*TL_17110*), *TcTPS15* (*TF_5958*), *TcTPS16* (*TF_2226*), and *TcTPS17* (*TL_10575*).

These transcript lengths ranged from 600 to 628 amino acids (Table 1), and the length distribution was close to that of known conifer sesquiterpenes and monoterpene synthases, compared with diterpene synthases. Sequence alignment with selective terpene synthase showed a conserved structure of the β-γ domain (Appendix A), which suggests their possible function as monoterpene or sesquiterpene synthases.

Transcript annotation against the NCBI database showed that *Tc*TPS13, *Tc*TPS15, and *Tc*TPS16 have similar sequences to the α-pinene synthase from *Chamaecyparis formosensis*. *Tc*TPS14 shares its highest similarity with the terpinolene synthase from *Chamaecyparis obtusa*, and *Tc*TPS17 has the best result against the previously identified monoterpene synthase from *T. cryptomerioides*. According to these results, the monoterpene syntheses sequences from Cupressaceae are closely related, and they might form a phylogenetic cluster, which was also observed in sesquiterpene and diterpene synthases.

To further investigate the genome structure of the identified TPSs, we attempted to amplify their genome sequences, and successfully obtained the sequences of *Tc*TPS14, *Tc*TPS15, and *Tc*TPS16. Compared with the ancestral form of terpene synthase (Figure 1, *Abies grandis* abietadiene synthase [26]), the *Tc*TPS14-16 has fewer members of exons due to possible intron loss events, but the structures share the same composition as *Abies grandis* monoterpene synthase (limonene synthase) [27]. The previously published intron XII loss was not observed in this study [19], thus it is likely an independent evolutionary event that occurred after the differentiation of sesquiterpene and monoterpene synthases.

### 2.2. Phylogenetic Analysis

The phylogeny was built by using collected terpene synthases from gymnosperm and angiosperm species, and *Physcomitrella patens ent*-copalyl pyrophosphate/ *ent*-kaurene synthase was used as the root of the tree due to its ancestral status. The first cluster branched from the root was diterpene synthases, and then second clusters were formed, which were mostly occupied by monoterpene and sesquiterpene synthases (Figure 2). This cluster was divided into several groups: tri-domain sesquiterpene synthases from the Pinaceae species; sesquiterpene synthases from Pinaceae/Cupressaceae; monoterpene/sesquiterpene synthases from the angiosperm species; and monoterpenes from the Pinaceae and Cupressaceae species (Figure 2). The phylogeny showed that the TPSs from Cupressaceae were clustered and formed a subgroup. This routine was observed not only in diterpene and sesquiterpene synthases but also in monoterpene synthases in an absolute way. Similar patterns suggested that conifer sesquiterpene and monoterpene synthases might have evolved from the same ancestor, and multiple evolution events led to a divergent biosynthetic network and diverse terpenoid structures, which would be beneficial for plants to dynamically adapt to ecological stresses [13,18,19].

*Tc*TPS13, *Tc*TPS15, and *Tc*TPS16 were closer to each other, and *C. formosensis* α-pinene synthase and *Tc*TPS14 and *TcTPS*17 were likely grouped with *C. obtusa* terpinpinolene synthase and *T. cryptomerioides* terpinolene/α-pinene synthases (Figure 2). The phylogenetic result was identical to the BLAST results conducted earlier. However, the function of monoterpenes is not phylogenetically conserved, and the diversity is limited by few functionally characterized enzymes. Thus, it is important to further understand the enzyme function of *Tc*TPS13~*Tc*TPS17.

### 2.3. In Vitro Enzymatic Assay

To understand the biosynthetic product of newly discovered *Tc*TPSs, heterologous protein expression, purification, and in vitro reaction were used. Because of the volatile properties of monoterpenoids, we used solid phase microextraction (SPME) to absorb the headspace of each reaction.

From the GC-MS analysis results, *Tc*TPS13 and *Tc*TPS14 shared the same product profile (Figure 3, RT 12.46/12.48 min), and geraniol was detected according to the comparison of spectral libraries and authentic standards. The product of *Tc*TPS15 had different retention times (Figure 4A, RT 8.44 min), and after comparison with standards, it had identical retention times and mass spectra to linalool. *Tc*TPS16, which was phylogenetically close to *C. formosensis* α-pinene synthase, shared the same metabolite profile as β-pinene (Figure 4B, 5.61 min). However, *Tc*TPS17 lacks a monoterpenoid product, although its expression can be observed by Western blot, and it may result from abnormal protein folding, which forms a functionless protein. As described above, *Tc*TPSs 13 and 14 are geraniol synthases, *Tc*TPS15 is linalool synthase, and *Tc*PS16 is β-pinene synthase.

The plasticity of the monoterpene synthase catalytic pocket was shaped by neighboring residues, and their location across helix D, F, G, H, J, and J/L loops (Figure 5), which can be grouped into four major plasticity regions identified via previous mutation assays [17]. For the biosynthesis of hydroxylated acrylic monoterpenoids, the intervention caused by water molecules in the reaction is the key. In limonene synthase from *Mentha spicata*, the lone side chain of M458 and H579 at region 3 and the J/K loop stabilize the cyclic intermediate, and the reaction was intervened by water once it was mutated to a less bulky amino acid, which formed hydroxylated acrylic monoterpene [15]. In *Tc*TPS13 and 15, the Val located at the M458-related location (Figure 5) indicates that a smaller amino acid could be essential to geraniol and linalool formation. However, the same Val was observed at M458 on *Tc*TPS 16 (α-pinene synthase); therefore, the key residues that determine the water interruption still need to be discovered.

On the other hand, from the comparison results of acyclic monoterpene synthase and bicyclic monoterpene synthase, we were able to identify Phe in *Tc*TPS13 and *Tc*TPS15 at the associated site of Y327 on *Tc*PinS (Figure 4), which could terminate the second cyclization of cations [21]. The conserved Tyr residues among conifer pinene synthases indicate that it is crucial to pinyl cation formation. On the other hand, some residues were characterized in determining the orientation of the pinyl cation from *Abies grandis* pinene synthase, and C372S and F579W preferred α-pinene formation instead of β-pinene [28]. However, the *T. cryptomerioides* pinene synthases exhibited seemingly opposite seanario: Cys on α-pinene synthase and Ser on β-pinene synthase, but the detailed mechanism needs to be verified in future experiments.

### 2.4. Characterized Monoterpene Biosynthetic Pathway in Taiwania cryptomerioides

Pinenes are major volatile terpenoids in *T. cryptomerioides*, and in addition to α-pinene and α-pinene/terpinolene synthase, we functionally characterized a β-pinene synthase (*Tc*TPS16) in this study, which allowed us to explore pinene biosynthesis in *T. cryptomerioides* (Figure 6) [7,9,11]. In *Tc*PinS and *T**c*TeoS mutagenesis studies, the slight exchange of residue would result in the ratio of monocyclic and bicyclic products [21]. Compared with *Tc*PinS and *Tc*TeoS, *Tc*TPS16 was closer to pinene synthase from *C. formosensis*, indicating that the phylogenetic relationship of Cupressaceae monoterpene synthases was dominated by species instead of enzymatic products.

Meanwhile, *Tc*TPS13, 14, and 15 were identified as geraniol and linalool synthases, although monoterpene alcohol was only present in trace amounts in extracts, volatiles, and essential oils of *T. cryptomeioides* needles and woods. In fact, monoterpene alcohols have been commonly observed in various plants, and their derivatives often exhibit excellent bioactivity that may be beneficial for plants to overcome biotic and abiotic stress [29,30,31]. The ecological and physiological functions of geraniol, linalool, and their unknown derivatives in *T. cryptomerioides* need to be further studied.

These results showed that transcriptome mining allowed us to discover novel transcripts for exploring specialized metabolism, and could be useful for further metabolite engineering and breeding.

## 3. Materials and Methods

### 3.1. Plant Material

*T. cryptomerioides* needles, cones, and seeds were collected from the Chi-Tou tract of the Experimental Forest at National Taiwan University and the Hui-Sun Forest Experimental Station at National Chung Hsing University. The samples were harvested from healthy 70-year-old trees, immediately frozen in liquid nitrogen and stored at −80 °C for further usage.

### 3.2. Transcriptome Analysis

Published *Taiwania* transcriptome libraries were used in this study, which included needles, cones, and wood [23]. To identify monoterpene synthase candidates, the previously established data were queried against monoterpene synthases in the NCBI/GenBank database using BLASTx searches.

### 3.3. RNA Extraction and TPS Isolation

Total RNA was isolated by using the pine tree method [32], and the RNA concentrations were then estimated by using a Nanodrop spectrophotometer (Thermo Scientific, Waltham, MA, USA). Each cDNA was synthesized from 1 μg of total RNA using SuperScript III reverse transcriptase (Invitrogen, Waltham, MA, USA). TPS candidates were amplified by using Phusion high fidelity DNA polymerase (Thermo Scientific) with specific primers. The full-length gene was obtained by conducting 5′ and 3′ rapid amplification of cDNA ends (RACE) PCR (Invitrogen). All amplicons were cloned into the pGEMTeasy vector (Promega, Madison, WI, USA) for further sequence verification. For heterologous expression in *Escherichia coli*, the open reading frame of the gene was subcloned into the pTYB12 (NEB, Ipswich, MA, USA), pET21, or pET28 expression vector (Merck, Darmstadt, Germany). If necessary, plastidial transit peptides were predicted and removed for subcloning. The protein expression of TPS identified in this study was verified by conducting polyacrylamide gel electrophoresis and Western blot. All primers used in this study are listed in Appendix A.

### 3.4. Genome Structural Analysis

Genomic DNA was isolated from needles of *T. cryptomerioides* using the Plant Genomic DNA Purification Kit (GeneMark, Taichung Taiwan) according to the manufacturer’s protocol. The genomic fragments of terpene synthase were amplified by using specific primers (Appendix A), and were subcloned into the pGEMTeasy vector for sequence verification.

### 3.5. In Vitro Enzyme Assays

The plasmid containing *Tc*TPS was transformed into *E. coli* C41 (DE3) cells (Lucigen, Middleton, WI, USA), and the transformed colonies were selected by suitable antibiotics. The transformed cells were grown in Luria–Bertani (LB) medium at 37 °C until the OD600 reached 0.6, and then isopropyl β-D-1-thiogalactopyranoside (IPTG) was added to a final concentration of 0.4 mM. The induced cultures were incubated at 16 °C. After 16~22 h of induction, the cells were collected by centrifugation. The cell pellets were lysed, and the recombinant proteins were purified using TALON Superflow purification resin (GE, Boston, MA, USA) or chitin resin (NEB), as previously described [23,24,25]. *All* enzymatic assays for monoterpene synthases were carried out by using 20 μL of GPP substrate (Sigma, St. Louis, MI, USA) and 50 μL of purified protein in a total volume of 500 μL (assay buffer: 50 mM 4-(2-hydroxyethyl)-1-piperazineethanesulfonic acid (HEPE), pH 7.2, 100 mM KCl, 10 μM MnCl_2_, 5% glycerol 2 mM DTT). The enzymatic reaction was sustained for 1 hr at 30°C and then extracted by solid phase microextraction (SPME) (carboxen-polydimethylsiloxane, 75 μm, Supelco, Bellefonte, PA, USA) after the SPME fiber was conditioned by heating at 250 °C for 15 min. The extract was placed at room temperature for 20 min. Once the fiber was removed from the headspace of the reaction, it was immediately desorbed at the GC injection port (250 °C) of a PolarisQ Ion Trap gas chromatogram/mass spectrometry system (Thermo Scientific) equipped with a DB-5 capillary column (Agilent Technologies, Santa Clara, CA, USA). The slope of the oven temperature was set as follows: 60 °C, increased by 5 °C/min up to 130 °C, increased by 30 °C/min up to 260 °C, and held for 5 min. Monoterpene standards were purchased (Sigma) and diluted to 200 ppm for GC-MS analysis, and reference mass spectral libraries from the National Institute of Standards and Technology (NIST) and Wiley were used in this study.

### 3.6. Phylogenetic Analysis

Selected TPS protein sequences were aligned by performing multiple sequence alignment (MUSCLE) [33], and the neighbor-joining method employed by MEGA X software [34] was used for phylogenetic analysis based on 100 bootstrap replications. The result was then illustrated by using FigTree V1.4.4 (http://tree.bio.ed.ac.uk/software/figtree/, accessed on 27 September 2021). All the sequences of phylogenetic construction are described in Appendix A.

### 3.7. Accession Number

The monoterpene sequences isolated in this study have been submitted to the National Center for Biotechnology Information (NCBI) under accession number QHZ00920.1~QHZ00924.1 (*T**c*TPS13~*Tc*TPS17). The published [23] transcriptome sequence (SRP062764) library was used.

## 4. Conclusions

In this study, a phylogenetically related cluster of Cupressaceae monoterpene synthases was identified. This discovery proved the distant phylogenetic relationship of Cupressaceae terpene synthases, which includes not only ancestral synthases (sesquiterpene and diterpene synthases) but also late evolved monoterpene synthases. By conducting in vitro enzymatic assays, we functionally characterized geraniol synthases (*Tc*TPS13 and *Tc*TPS14), linalol synthase (*Tc*TPS15) and β-pinene synthase (*Tc*TPS16). These results illustrated monoterpene biosynthesis in *T. cryptomerioides* and will be a valuable foundation to understand the ecological role of diverse monoterpenoids and will be beneficial for molecular breeding and forest management.

## Figures and Tables

**Figure 1 plants-10-02404-f001:**
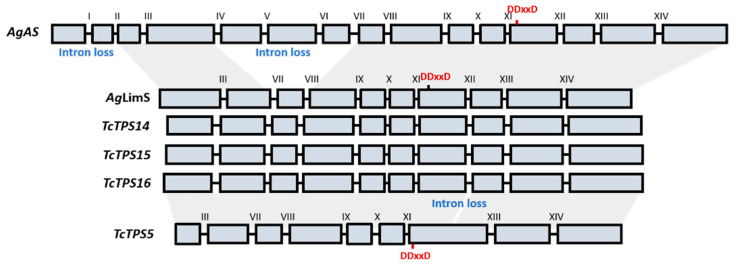
The proposed evolution of genome structure of conifer terpene synthase. *AgAS*: *Abies grandis* abietadiene synthase [26]; *AgLimS*: *A. glauca* limonene synthase [27]; *TcTPS5*: *T. cryptomerioides* longifolene synthase [25]. The genome structures were illustrated based on published research [19], and the location of the active motif (DDxxD) was marked.

**Figure 2 plants-10-02404-f002:**
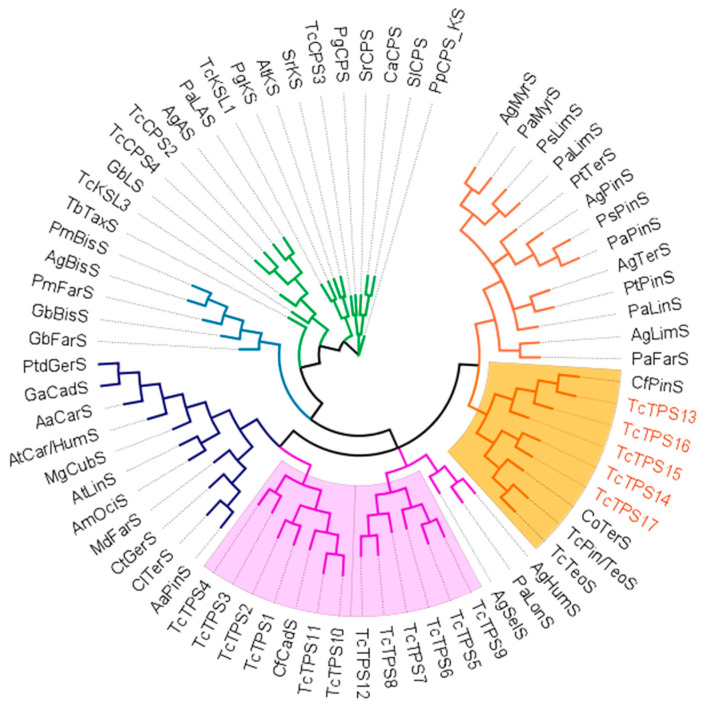
Phylogenetic analysis of selected terpene synthases. Each branch was colored according to its function and species orientation. Green: diterpene synthases; blue: tri-domain sesquiterpene from gymnosperms, deep blue: mono-/sesquiterpene synthases from angiosperms; pink: sesquiterpene from gymnosperms; orange: monoterpene synthases from gymnosperms. The Cupressaceae-associated subgroups are marked with a colored background.

**Figure 3 plants-10-02404-f003:**
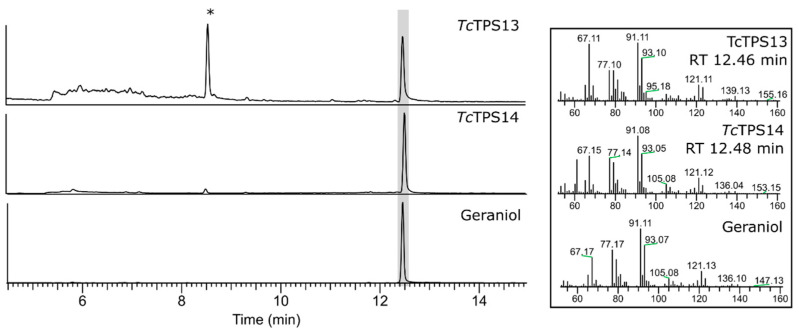
GC-MS analysis of the enzymatic products of *Tc*TPS13 and *Tc*TPS14 and comparison with the geraniol standard.

**Figure 4 plants-10-02404-f004:**
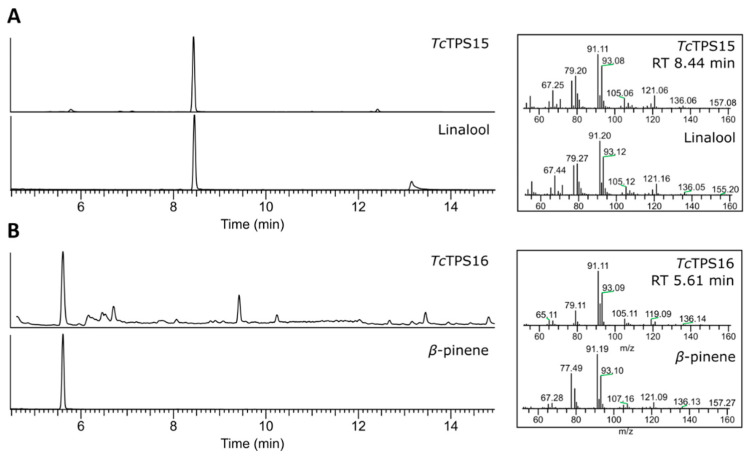
GC-MS analysis of the enzymatic products of *Tc*TPS15 and 16. (**A**) The metabolic profile of the *Tc*TPS15 product and linalool standard; (**B**) the metabolic profile of *Tc*TPS16 and β-pinene standard.

**Figure 5 plants-10-02404-f005:**
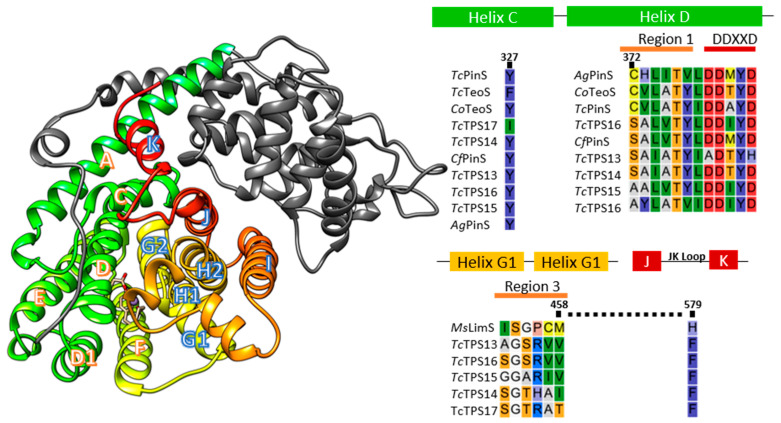
The protein structure of *Tc*TPS13 (model template ID: 2onh.1. A) and catalytic residue alignment. Helixes A to K and their related locations are illustrated in the left panel. The full alignment is illustrated in Appendix A.

**Figure 6 plants-10-02404-f006:**
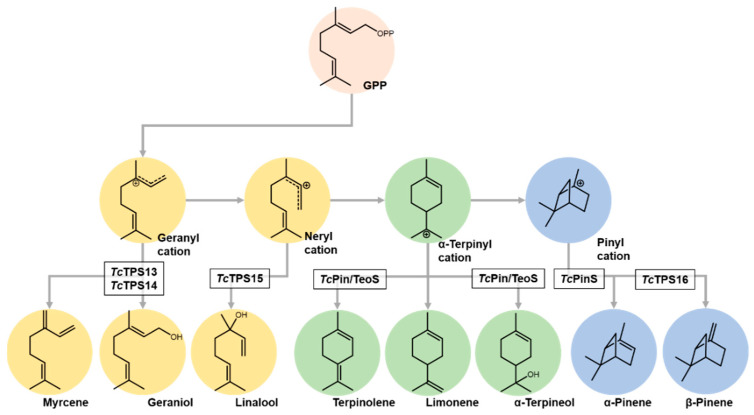
Functionally characterized monoterpene biosynthetic pathway.

**Table 1 plants-10-02404-t001:** Summary of *T. cryptomerioides* monoterpene synthases identified in this study.

Sequence ID	Gene Name	Coding Region (a.a.)	Best Blast Hit ^1^
TF_29358	*TcTPS13*	628	α-Pinene synthase [*Chamaecyparis formosensis*] ^2^
TL_17110	*TcTPS14*	607	Terpinolene synthase [*Chamaecyparis obtusa*] ^3^
TF_5958	*Tc* *TPS15*	610	α-Pinene synthase [*Chamaecyparis formosensis*] ^2^
TF_2226	*Tc* *TPS16*	617	α-Pinene synthase [*Chamaecyparis formosensis*] ^2^
TL_10575	*Tc* *TPS17*	593	Terpene synthase [*Taiwania cryptomerioides*] ^4^

^1.^ All blast results’ E values are 0.00,.^2.^ Accession number: ABW80964.1,.^3.^ Accession number: BAI53108.1,.^4.^ Accession number: AIO10963.1.

## Data Availability

The data presented in this study are available in article and Appendix A.

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
