# Peer review of "Unveiling Monoterpene Biosynthesis in Taiwania cryptomerioides via Functional Characterization"

_plants, 2021, doi:10.3390/plants10112404_

Round 1

Reviewer 1 Report

Dear Authors, 

Please see my comments below - 

line 10 - synthase

line 14 - delete the (those)

line 66 - might be caused by...

line 78- capitalize first letter

line 92- pinene capitalized and in the text below there are also a few capitalisation mistakes.

line 99 - to...to

Please check and change the alpha symbol, it looks odd in the text - use the alpha symbol from built-in character library of MSword.

line 127 - indetified (check english)

line 138- synthase.....synthase

line 146 - on the other hands....?

line 219 - ppm with small caps

I think the manuscript would benefit if the M&M section would be expanded also into the results. It is difficult to understand what was done in the Results section and why. The figures should also be better placed - I suggest you place them right after the paragraph describing the specific figure. Overall the manuscript is interesting but it lacks depth in my opinion. There are more important results that culd be presented as figures, rather than chromatograms of the volatiles (which are surprisingly few).

Also, there are mistakes in the correct use of gramatically correct English language, please have a native speaker go throuh the manuscript and correct the mistakes or perhaps you could use the spell checking services provided by publisher.

Author Response

Thank you for the detailed correction, and all the mistakes you mentioned in last response were fixed. The writing was checked and improved by outsourcing edited service as reviewer suggested.

The figures have been placed right after the paragraph. For better understanding, the logic of each experiments has been described in each section. We also included the description of the TPS evolution in section 2.1 via comparison of genomic structure of selective conifer TPSs, and we hope this would be helpful in understanding the logic behind the divergence of terpene synthases and the necessity of functional characterization of terpene synthases in non-model species.

Reviewer 2 Report

The research is interesting in the area of the valuation of monoterpene biosynthesis in Taiwania cryptomerioides via functional characterization.. Nevertheless, the manuscript needs to be improved in order to provide enough information to justify its importance and novelty. Please re-structure in order to explain the novelty of the research, the research question, hypothesis and the objectives. Please use the section background to explain the importance and novelty of this research and its objective. Please conclude according to the main objectives of the research.
Please include in section Conclusions more practical applications of your findings.
In this form the objective of the study is actually the conclusions????

Author Response

Thank you for the valuable advice. We have revised the writing, especially for introduction and conclusion to emphasize the novelty and importance of this study.

Reviewer 3 Report

The results are valuable, well presented and the paper could be accepted in the present form.

Author Response

Thank you for your agreement.

Round 2

Reviewer 2 Report

The manuscript has been improved and my recommendation is to be published.